# Tumor Microenvironment Features as Predictive Biomarkers of Response to Immune Checkpoint Inhibitors (ICI) in Metastatic Clear Cell Renal Cell Carcinoma (mccRCC)

**DOI:** 10.3390/cancers13020231

**Published:** 2021-01-10

**Authors:** Audrey Simonaggio, Nicolas Epaillard, Cédric Pobel, Marco Moreira, Stéphane Oudard, Yann-Alexandre Vano

**Affiliations:** 1Medical Oncology, Hôpital Européen Georges Pompidou, APHP Centre–Université de Paris, 75015 Paris, France; audrey.simonaggio@aphp.fr (A.S.); nicolas.epaillard@gustaveroussy.fr (N.E.); cedric.pobel@gustaveroussy.fr (C.P.); stephane.oudard@aphp.Fr (S.O.); 2INSERM, UMR_S 1138, Centre de Recherche des Cordeliers, Team “Cancer, Immune Control and Escape”, University Paris Descartes Paris 5, Sorbonne Paris Cite, 75006 Paris, France; marco.moreira@inserm.fr; 3INSERM UMR-S1147, Université de Paris, Sorbonne Université, 75006 Paris, France

**Keywords:** clear cell renal cell carcinoma, immune checkpoint inhibitors, biomarker, genomic signature, transcriptomic analysis

## Abstract

**Simple Summary:**

In recent years, the therapeutic armamentarium of mccRCC has changed dramatically with the emergence of targeted therapy and immune checkpoint inhibitors, used alone or as a combination. However, mccRCC still have a poor prognosis and a significant portion of patients experience primary or secondary resistance. The tumor microenvironment plays a major role in promoting tumor resistances. This review aims (i) to provide an overview of the components of the RCC tumor microenvironment, (ii) to discuss their role in disease progression and resistance to ICI, (iii) to highlight the current and future ICI predictive biomarkers assessed in mcccRCC.

**Abstract:**

Renal cell carcinoma (RCC) is the seventh most frequently diagnosed malignancy with an increasing incidence in developed countries. Despite a greater understanding of the cancer biology, which has led to an increase of therapeutic options, metastatic clear cell renal cell carcinoma (mccRCC) still have a poor prognosis with a median five-years survival rate lower than 10%. The standard of care for mccRCC has changed dramatically over the past decades with the emergence of new treatments: anti-VEGFR tyrosine kinase inhibitors, mTOR Inhibitors and immune checkpoint inhibitors (ICI) such as anti-Programmed cell-Death 1 (PD-1) and anti-anti-Programmed Death Ligand-1 (PD-L1) used as monotherapy or as a combination with anti CTLA-4 or anti angiogenic therapies. In the face of these rising therapeutic options, the question of the therapeutic sequences is crucial. Predictive biomarkers are urgently required to provide a personalized treatment for each patient. Disappointingly, the usual ICI biomarkers, PD-L1 expression and Tumor Mutational Burden, approved in melanoma or non-small cell lung cancer (NSCLC) have failed to distinguish good and poor mccRCC responders to ICI. The tumor microenvironment is known to be involved in ICI response. Innovative technologies can be used to explore the immune contexture of tumors and to find predictive and prognostic biomarkers. Recent comprehensive molecular characterization of RCC has led to the development of robust genomic signatures, which could be used as predictive biomarkers. This review will provide an overview of the components of the RCC tumor microenvironment and discuss their role in disease progression and resistance to ICI. We will then highlight the current and future ICI predictive biomarkers assessed in mccRCC with a major focus on immunohistochemistry markers and genomic signatures.

## 1. Introduction

Kidney cancer accounts for 3 to 5% of all malignancies, with an increasing incidence in developed countries. In 2018, about 330,000 new cases were diagnosed and 120,000 patients died from kidney cancers [1]. A greater understanding of the molecular and immune tumor characteristics led to a rising of therapeutic options. Until the 2000s, therapeutic options were limited, based on cytokine therapies: interleukine-2 (IL-2) and interferon alpha (IFN-a). Objective response rates were low (5–20%), and the safety profile was often limiting because of cardiac and respiratory toxic effects. Interferon alpha gave an improvement in one-year survival of 12% and in median survival of 2.5 months versus medroxyprogesterone acetate [2,3,4]. During the 2000s, new techniques of genomic and phenotypic analysis allowed a greater understanding of clear cell renal cell carcinoma (ccRCC) biology. VEGFR (Vascular Endothelial Growth Factor Receptor) and PI3K/AKT/mTOR (Phosphatidyl-Inositol-3′-Kinase/Protein Kinase B/Mammalian Target Of Rapamycin) emerged as two major pathways involved in ccRCC carcinogenesis.

Consequently, during the 2000s, anti-VEGFR TKI and mTOR inhibitors emerged as the new standards of care for metastatic clear cell renal cell carcinoma (mccRCC). Sunitinib [5] and pazopanib [6] were approved for the untreated Memorial Sloan Kettering Cancer Center (MSKCC) favorable and intermediate group. Temsirolimus and everolimus [7] were approved for first-line MSKCC intermediate and poor risk mccRCC patients and after anti-VEGFR TKI failure. Compared with the placebo or interferon, sunitinib and pazopanib provided a survival benefit with a median overall survival of 29.3 and 28.4 months, respectively [8].

In the last few years, the standard of care for mccRCC has changed dramatically with the emergence of immune checkpoint inhibitors (ICI) anti-Programmed cell-Death 1 (PD-1) and anti-Programmed Death Ligand-1 (PD-L1) used as monotherapy or as a combination with anti-cytotoxic T-lymphocyte-associated antigen 4 (CTLA-4) or antiangiogenic.

According to the last ESMO guidelines update, mccRCC front-line therapeutic options remain guided by the International Metastatic RCC Database consortium (IMDC) Risk Score. The combination of pembrolizumab plus axitinib is recommended irrespective of the IMDC prognostic subgroups and PD-L1 biomarker status, while the combination of nivolumab plus ipilimumab is restricted to patients with intermediate and poor-risk disease. The VEGFR tyrosine kinase inhibitor (TKI) is now only recommended in case of contraindication to the above-mentioned combinations [9]. Survival benefits offered with these new combinations will be extensively described in Section 4.

Although ICI combinations have changed the prognostic of mccRCC with impressive overall response rates (ORR) (respectively, 55% with pembrolizumab-axitinib [10] and 42% with nivolumab-ipilimumab [11]), many patients do not respond to immunotherapy, reflecting a primary resistance to ICI. Durable responses remain scarce, and secondary resistance rates approach 100%. Factors contributing to primary or acquired resistance are manifold, including patient-intrinsic factors and tumor cell and tumor microenvironment-intrinsic factors. In this review, we will describe the Tim-3 Expression (TME) composition in renal cell carcinoma, focusing on the vascular, the stromal and the immune compartments. We will make a particular focus on the potentiality of the TME to induce resistance to ICI and highlight the current and future ICI predictive biomarkers assessed in mccRCC.

## 2. Tumor Microenvironment: Definition and Available Study Methods

### 2.1. Definition

The TME is a spatially organized and dynamic network composed both of tumor cells, immune cells, endothelial cells, structural molecules, extra cellular matrix and many other cells as neuroendocrine cells, adipose cells or stromal cells [12]. This ecosystem modulates all aspects of tumor development, tumor progression and therapy resistance [13]. In 2017, applying mass cytometry for the high-dimensional single-cell analysis of kidney primary tumors, Chevrier et al. depicted an in-depth Immune Atlas of Clear Cell Renal Cell Carcinoma [14]. T cells were the main immune cells population (almost 50%). Mean frequencies of the myeloid cells, natural killer cells (NK cells) and B cells were 31%, 9% and 4%. Only a few granulocytes and plasma cells were identified [14].

### 2.2. Study Methods

#### 2.2.1. Immunohistochemistry (IHC) and Scoring

An immunoscore is used to decipher the TME IHC on a tumor section using analysis software allowing the estimation of immune cells densities in the center of the tumor and in the invasive margin. Densities of CD3+ and CD8+ cells are calculated for both regions. The evaluation of an immunoscore is based on the obtention of a score between 0 (I0), indicating a low density of the two immune cells populations in both cores, to 4 (I4), reflecting a high density of the two types in both cores [15]. The correlation between the immunoscore and survival outcomes was first validated in colorectal cancer [15,16]. In a recent study focusing on ccRCC, a favorable immunoscore was associated with improved survival outcomes [17].

One of the major limitations of IHC analysis is the limited number of markers using bright field IHC. Solutions are available to increase the number of markers assessed on one slide using fluorescent based multiplex, extending the number of markers up to eight [18,19]. Antibody DNA barcoding (InSituPlex^®^ Technology by ULTIVUE for 12-plex or CODEX^®^ by AKOYA Bioscience for a 29-plex maximum) is a new method of cancer cell profiling using DNA-conjugated antibodies. The DNA-conjugated antibodies contain a photo-cleavable linker that enables their release after exposure to ultraviolet light. The transmitted signal is then measured and translated to protein expression. It enables to increase the number of available markers. For RCC, IHC remains the gold standard to categorize the diverse subtype of renal tumors and to assess the expression of immune checkpoints such as PD1, PDL1, lymphocyte-activation gene 3 (LAG3) or Tim-3. The predictive and prognostic values of these biomarkers will be discussed later in this review.

#### 2.2.2. Flow Cytometry

Cytometry allows a larger set of analysis but does not provide spatial information. Flow cytometry based on the use of fluorescent coupled antibodies can go up to sixteen markers. The overlap of the signals is a major limitation of this tool. The high-dimensional mass cytometry (CyTOF cytometry by time-of-flight) involving rare earth metal-coupled antibodies is a new promising TME study method. The CyTOF allows to analyze more than 50 markers on a single cell. Recently, mass cytometry application has produced impressive results when coupled to single cell RNAseq, resulting in atlases, as demonstrated in ccRCC by Chevrier et al. [14]. Such methods may enable to investigate the expression of immunotherapeutic targets on peripheral immune blood cells. It should be noticed that they require large amounts of fresh tissue, which remains a major limitation.

#### 2.2.3. Transcriptomic Data and Deconvolution Tools

The Microenvironment Cell Population counter (MCP counter) is a software that uses transcriptomic data and deconvolution tools to “decipher the contribution of different cell populations to the overall transcriptomic signal of heterogeneous tissue samples” [20]. Deconvolution is an algorithm-based method used to deduce the abundance of cell types and cell expression into an heterogeneous sample using gene expression data [20]. MCP counter allows the quantification of the absolute abundance of 10 distinct populations (eight immune cell types, including T cells, CD8+ T cells, NK cells, cytotoxic lymphocytes, B cell lineage, monocytic lineage cells, myeloid dendritic cells and neutrophils, and two nonimmune stromal populations, including endothelial cells and fibroblasts [20].

MCP counter allows inter-sample comparisons of immune and stromal cells and has been used for ccRCC analysis [21]. Those signatures have been used with the Gene Set Enrichment Analysis (GSEA), allowing the definition of the “immunophenoscore” in association with immunotherapies response in ccRCC, melanoma and pan-cancer. For example, Senbabaoglu et al. used a gene expression-based computational method to characterize the infiltration levels of 24 immune cell populations (by interrogating expression levels of genes) in 19 cancer types. Three groups of ccRCC were identified: T cell-enriched (15.7%), heterogeneously infiltrated (61.9%) and noninfiltrated (22.4%). CcRCC was the most highly T cell-infiltrated tumor type. Focusing on a small sample of patients (*n* = 6), the major histocompatibility complex (MHC) class I antigen presenting machinery expression and T-cell infiltration were elevated in patients with a partial or complete response to nivolumab [22,23].

In the last 15 years, many new “omics” technologies have been permitted to obtain high-resolution data and unprecedented views of the biological and cancer systems. This has led to the development of predictive genomic signatures. In their review, Sung et al. summarized the concept of molecular signature as “ a set of biomolecular features (DNA sequence, DNA copy number, protein…) together with a predefined computational procedure (using supervised or unsupervised classification) that applies those features to predict a phenotype of clinical interest ” [24]

Some genomic signatures are used in routine practice, such as the Oncotype Dx signature in breast cancer [25,26]. Genomic signatures may play such an important role in other malignancy, especially mccRCC.

The main characteristics, the strengths and the weaknesses of the TME study methods, are summarized in Figure 1 [27,28,29,30,31].

## 3. TME Components as Predictors of Systemic Treatment Efficacy

Major cross-talks between the vascular, the immune and the stromal compartments are summarized in Figure 2.

### 3.1. Vascular Compartment

#### Endothelial Cells and Hypoxia

RCC is one of the most hyper-vascularized tumors, composed of disorganized vessels. Due to this disorganization, nutrients and oxygen intakes are insufficient, leading to hypoxia and a lower pH tumor, which both contribute to tumor progression [32]. Hypoxia induces the upregulation of different genes involved in glucose metabolism, cell angiogenesis, cell proliferation, the polarization of macrophages into tumor-associated macrophages (TAM) and regulatory T cells (T-regs) recruitment and infiltration of myeloid-derived suppressive cells (MDSCs), leading to the inhibition of CD3+ T cells and cytotoxic functions of CD8+ T cells [33]. Consequently, the release of hypoxia-induced factor 1a and 2a (HIF-1a and HIF-2a) induces an increasing expression of PD-L1 in tumor cells [34,35].

The high levels of HIF-1 and HIF-2 mediate the generation of vascular endothelial growth factor (VEGF), explaining the high vascularization of RCC. VEGF acts as an escape pathway to immunosurveillance by increasing immune checkpoints as CTLA-4, T-cell immunoglobulin and mucin domain-containing protein-3 (TIM3) and lymphocyte-activation gene 3 (LAG3) on T-cell surfaces and PD-L1 on dendritic cells [36]. VEGF also promotes the recruitment of Treg cells and MDSCs and suppresses the maturation of dendritic cells [37]. Hypoxic tissue also demonstrates adenosine deposit generated by the CD39-CD37 system and acting as an immune escape pathway by suppressing the effector effect of T cells [37]. Hypoxia modifies the intra cellular concentration of lactate, turning the macrophage phenotype into type 2 polarization. It also increases the expression of survival and migration genes in tumor cells and the production of molecules inhibiting natural killers T cells (NK), dendritic cells (DC) and T-cell cytotoxicity [37]. Altogether, these compelling data demonstrated that hypoxia induced by disorganized vessels is involved in the antitumor immune response, which is the basis of the rationale to combine antiangiogenic therapies and ICI.

As described above, renal cell carcinoma is associated with a hyper angiogenic state related to an overexpression of VEGF and other angiogenic-related genes, such as ESM1, PECAM1 or FLT1. Using transcriptomic analysis and computational procedure, the expression of theses angiogenesis-related genes was assessed both in adjuvant and metastatic settings. The main results of these mRNA analyses will be discussed within this review.

### 3.2. Immune Compartment

#### 3.2.1. CD8+ T Cells

Fridman et al. recently reviewed the role of CD8+ T cells in cancer prognosis and treatment. Although the density of CD8+ T cells in tumors is associated with a good prognosis in most cancer types, including breast, hepatocellular, colorectal, melanoma, bladder, lung and head and neck cancers, CD8+ T cell infiltration is associated with a worse prognosis for RCC, follicular lymphoma, Hodgkin lymphoma and prostate cancers [38]. Focusing on RCC, this negative correlation could be explained in part by the high levels of CTLA and PD-1 expression on T cells [23]. In 2015, Giraldo et al. identified two groups of ccRCC tumors with high CD8+ T-cell infiltrates and opposite survival outcomes. The first group was associated with good prognosis and characterized by a high expression of immune checkpoints and the absence of mature dendritic cells. The second group was associated with a worse prognosis and was characterized by a low expression of immune checkpoints and the presence of mature peritumoral dendritic cells. One hypothesis to explain this observation may be that CD8+ T cells are often exhausted in ccRCC and that CD8+ T cells could only be educated in presence of a high density of mature dendritic cells located in tertiary lymphoid structures. Both immune checkpoints and dendritic cell (DC) localization in the tumor microenvironment modulate the clinical impact of CD8+ T cells in ccRCC. The increased angiogenesis level observed in RCC could result in a low density of tertiary lymphoid structures (TLS) and high number of immature DC located outside the TLS, leading to immature CD8+ T cells without cytotoxic capacity [39].

Besides the CD8+ T-cell density, the activated or inhibited status is a key predictor of treatment efficacy. Focusing on a cohort of 40 RCC patients, three dominant immune profiles were identified: (i) immune-regulated, characterized by polyclonal/poorly cytotoxic CD8+PD-1+Tim-3+Lag-3+Tumor-Infiltrated-Lymphocytes (TILs) and CD4+ICOS+ cells with a T-reg phenotype (CD25+CD1 27-Fox+3+/Helios+GITR+), (ii) immune-activated, enriched in oligoclonal/cytotoxic CD8+PD-1+Tim-3+TILs and (iii) immune-silent, enriched in TILs exhibiting the Renal-Infiltrated-Lymphocyte (RIL)-like phenotype. Immune-regulated tumors with a CD8+PD-1+Tim-3+ and CD4+ICOS+ PBL phenotypic signature displayed aggressive histologic features and a high risk of disease progression. These findings support that patients with immune-regulated tumors infiltrated with exhausted CD8+PD-1+Tim-3+Lag-3+ TILs could benefit from ICI alone or in combination and closer clinical follow-up [30].

LAG-3 is an immune checkpoint identified on exhausted CD8 T cells [30]. As an intensively studied biomarker through several clinical trials [40], it could also be an ICI target in mccRCC. A phase I study assessed IMP321, a recombinant soluble LAG-3-immunoglobulin fusion protein that agonizes MHC II driven dendritic cell activation and shows acceptable toxicity in patients with advanced RCC. Seven of eight evaluable patients treated at the higher doses experienced a stable disease at three months. No objective response was reported in this study [41]. FRACTION-RCC is an ongoing phase II trial testing various treatments for advanced RCC, including Relatlimab, an anti-LAG-3 antibody (NCT02996110).

TIM-3, another inhibitory co-stimulation molecule, could be a promising predictive biomarker in mccRCC. Granier et al. showed that the co-expression of TIM-3 and PD-1 on CD8 T cells was correlated with a higher TNM stage, larger tumor size and lower progression-free survival (PFS) in mccRCC [42]. This observation was confirmed by Ficial et al. in the last ASCO 2020 Virtual Annual Meeting [43]. By analyzing tumor tissue from the checkmate 025 study, they revealed that the high density of CD8+PD1+TIM3-LAG3-tumor infiltrating cells (TIC) was associated with better survival outcomes for patients receiving nivolumab: ORR (45.8% versus 19.6%, *p* = 0.01), clinical benefit (33.3% versus 14.1%, *p* = 0.03) and longer median PFS (9.6 months versus 3.7 months, *p* = 0.03) for an optimized cutoff. Such an association was not observed in the control everolimus arm. High levels of CD8+ PD1+ TIM3- LAG3- TIC were associated with an activation of the inflammatory response. Interestingly, the combination of PD-L1 expression on tumor cells with the density of CD8+ PD1+ TIM3- LAG3- TIC improved the predictive value, confirming previous results from Pignon et al. [43,44].

This would support a therapeutic strategy targeting TIM-3 alone or in combination with anti-PD-1/PD-L1. Several studies evaluating an anti-TIM-3 antibody are ongoing in different tumor types, such as Sym023, TSR-022 and MBG453, for patients with advanced solid tumor and/or hematologic malignancies (NCT 03489343, NCT 02817633, NCT02608268, NCT03066648, NCT03680508 and NCT03961971). Yet, there is no clinical trial focusing on mccRCC.

#### 3.2.2. Tumor-Associated Macrophage (TAM)

Macrophages are among the most abundant cells in the TME and are present at all stages of tumor progression. They mostly adopt a protumor phenotype in vivo both in primary and metastatic sites [45]. They can be divided into two main types: M1 type producing inflammatory cytokines: IL-12, IL-23 and IL-6 and M2 type expressing PD-1 ligands and producing anti-inflammatory cytokines: IL-10, TGF-β and IL-23. M2 macrophages also induce T-reg proliferation. Although TAM have been extensively studied in the last years, the question of their origin remains controversial. They are not solely derived from bone marrow has known before. To the last knowledges, they arise from yolk sac progenitors [46].

Focusing on RCC, the presence of extensive TAMs infiltration in the TME contributes to cancer progression and metastasis by stimulating angiogenesis, tumor growth, cellular migration and invasion, as well as recruitment of T-reg cells to the tumor site by secreting CCL20 or CCL22 [47]. Therapeutics strategies have been proposed to suppress TAM recruitment, to deplete their number, to switch TAMs into antitumor M1 phenotype and to inhibit TAM-associated molecules [48]. Moreover, TAM isolated from RCC induce the expression of CTLA4 and Foxp3 in CD4+ T cells [49], suggesting a possible therapeutic combination on TAM inhibition or TAM repolarization via CSF1R inhibitors and ICI [50]. A phase I clinical trial achieved a partial response rate of 30% and disease control of 100% with the anti-VEGFR, PDFGR and CSF1R tyrosine kinase inhibitor CM082 (X-82) in combination with everolimus for the treatment of metastatic RCC [51].

It should be noted that Martin Voss et al. reported recently that M2 macrophages, which were the most abundant infiltrating cell types, were associated with durable clinical benefit from anti-PD-1 therapy (*p* < 0.001). No association was found with TKI (*p* = 0.15) [52]. In the Nivoren ancillary cohort, CD163 (M2 macrophages) with higher density in the invasive margin was associated with better PFS (hazard ratio (HR) = 0.69, *p* = 0.016) but not overall survival (OS) (*p* = 0.5) in patients treated with nivolumab.

#### 3.2.3. Regulatory T Cells (Tregs)

T-regs are an immunosuppressive subset of CD4+ T cells characterized by the expression of Forkhead box protein P3 (FoxP3) [53]. They have both deleterious and beneficial effects. They regulate the immune system activity and maintain peripheral self-tolerance on the one hand and limit anticancer immunity in the other hand.

T-regs exert their immunosuppressive functions through various cellular and hormonal mechanisms. Vignali et al. summarized the main T-reg-suppressive mechanisms as follows: “suppression by inhibitory cytokines IL-10, TGF-β, IL-35, suppression by cytolysis (via granzyme A, granzyme B and perforine); suppression by metabolic disruption and suppression by modulation of dendritic-cell maturation or function (via indoleamine 2,3-dioxygenase (IDO) release and LAG3 binding to MHC class II molecules)” [54].

With regard to the RCC, the analysis of the immune infiltration in RCC from TCGA showed a higher proportion of Tregs in patients with a worse outcome [55].

From a therapeutic standpoint, the combination of everolimus and low-dose cyclophosphamide was evaluated in a phase I clinical trial in mccRCC patients. Cyclophosphamide, an alkylating drug, is known to deplete T-regs. Everolimus, an mTOR inhibitor, is known to control the expression of FoxP3 and, thus, to regulate T-regs. The primary objective was to evaluate the immune-modulating effects of different dosages of this combination, with the goal to achieve selective T-reg depletion. The combination of cyclophosphamide (50 mg once-daily) and a standard dose of everolimus (10mg once-daily) led to a reduction of T-regs and myeloid-derived suppressor cells. The combination therapy is evaluated in a phase II clinical trial [56], but the recent approval of new IO-based (immuno-oncology) combinations, as well as new TKI such as cabozantinib, lowered the interest of any everolimus-based combinations.

#### 3.2.4. B cells and Tertiary Lymphoid Structures (TLS)

After a period without breakthroughs of B cells in the immuno-oncology field, the importance and predictive role of B cells are now well-known. B cells are multifaceted cells, as both anti- and protumor roles have been reported, depending on their location in mature or immature tertiary lymphoid structures (TLS). In a recent review, Bruno et al. hypothesized that, in immature TLS, B cells release inhibitory factors, whereas, in mature TLS, they release antibodies and activate T cells [57].

Some activated B cells are characterized by a strong memory response again tumor-associated antigens (TAAs) and can release antibodies against tumor cells, leading to antibody-dependent cell death (ADCC). They are also required for T-cell activation [57,58,59]. However, B cells can act as paracrine mediators of solid tumor development by regulating diverse T lymphocyte responses through the release of several protumorigenic cytokines, such as IL-6, IL10, TNF-a or granulocyte monocyte-colony-stimulating factor (GM-CSF) [60]. In 2015, a particular subpopulation of B cells was designated as “B cells with a regulatory role” (B-regs), characterized as immunosuppressive cells secreting IL-10, IL-35 and TGF-β, triggering T-cell differentiation into T-regs to support immunological tolerance and inhibiting DC, CD8+ T cells and Th1 and Th17 lymphocytes. The differentiation of B-reg cells could be induced by a proinflammatory tumor environment [59,61,62].

The MCP counter analysis of tumor samples showed higher B cell-related genes in responder as compared to nonresponder patients in melanoma and ccRCC [62,63]. In sarcoma, patient clusters (SIC E) characterized by a high plasma cell signatures demonstrated an improved prognosis when treated with ICI anti-PD-1 [64]. In both of these previous studies published in Nature in 2020, the presence of B cells in tumors among TLS was associated with a favorable ICI response [57]. TLS appear as predictive biomarker of the ICI response.

TLS can be described as ectopic lymphoid formations localized in inflamed, infected or tumoral tissues. They constitute lymph node-like structures with a T-cell zone with mature dendritic cells and a follicular zone enriched in B cells proliferating and differentiating in the germinal center. These structures are associated with a good prognostic in patients with non-small cell lung carcinoma (NSCLC), colorectal cancer, breast, head and neck, pancreatic or gastric cancers, RCC or melanomas [38,65,66]. To focus on the clinical implications, therapeutics to enhance B-cell responses and TLS formations could be considered as a new combination therapy with ICI [57].

### 3.3. Stromal Compartment

#### 3.3.1. Myeloid-Derived Suppressor Cells (MDSCs)

The term “MDSCs” was first introduced in the scientific literature ten years ago. This heterogenous group of cells is made the of pathologic state of activation of monocyte and immature neutrophils. Basically, MDSCs consist of two major groups of cells: granulocytic or polymorphonuclear (PMN-MDSC) and monocytic (M-MDSC). MDSCs have pleiotropic effects. Focusing on cancer, the MDSC immunosuppressive role has been extensively studied. Faced with a weak and long duration activation signal, such as cancer, pathologic myeloid cell activation occurs, being mediated by soluble factors such as IL-6, IL-10, IL-1b, IFN-gamma and damage-associated molecular pattern (DAMP). This activation leads to the alteration of phagocytosis activity, to the release of reactive oxygen species (ROS), nitric oxide (NO), Prostaglandin E2 (PGE2), arginase I and anti-inflammatory cytokines. This results in the inhibition of the adaptive immunity and the promotion of tumor progression and metastasis. MDSCs inhibit antitumor activities of T and NK cells and stimulate T-regs [67,68,69,70,71]. Positive correlations have been reported between MDSC numbers in the peripheral blood and cancer stage in many types of cancers, including RCC [72]. Focusing on RCC, a positive correlation was observed between peripheral PMN-MDSC and tumor grade, suggesting a prognostic value [73]. In vitro studies have reported that the histone deacetylase inhibitor (HDACi) may influence MDSCs to a more differentiated status of macrophage or dendritic cells [74,75]. Using syngeneic mouse models of lung and renal cells, Orilliion et al. observed that the HDACi entinostat improves the antitumor effects of anti-PD1 in both mouse tumor models. The tumor growth was reduced, and survival was increased. The analysis of the MDSC response to entinostat showed a significant reduction of arginase-1, NOS level and protumorigenic cytokines, suggesting a modulation of the immunosuppressive TME [76]. Faced with these promising results, phase I/II clinical trials were initiated. One of them, an ongoing trial (NCT03024437), is assessing the safety and efficacy of atezolizumab in combination with entinostat and bevacizumab in patients with advanced RCC.

#### 3.3.2. Cancer-Associated Fibroblasts (CAFs)

The main function of the stromal compartment is to provide a functionally supportive tissue to epithelial cells and organs consisting of connective tissues and blood vessels. CAFs are the major component of this compartment. CAFS are a subpopulation of fibroblasts with a myofibroblastic phenotypes and are characterized by carcinogenic processes and fibrotic disorders [77]. They can be activated by growth factors released by tumor cells and differ from normal fibroblasts by an increase collagen and matrix protein production, an increase release of protumor factors and the expression of CAFs markers, including alpha-smooth muscle actin (a-SMA) and fibroblast activation protein (FAP) [12,78,79,80].

Among their functions, CAFs are able to stimulate the angiogenesis and promote tumor growth by releasing growth factors such as VEGF, platelet-derived growth factor (PDGF), transforming growth factor β (TGF-β), platelet epidermal growth factor (EGF) and fibroblast growth factor (FGF) [12]. The secretion of immunosuppressive substances such as interleukine-6 (IL-6) or indoleamine 2,3-dioxygenase (IDO) favor immune escape by promoting MDSCs M2-TAMs [81]. Among the growth factors secreted by CAFs, TGF-β plays a major role with pleiotropic protumorigenic effects. It promotes M2-TAM polarization, proinflammatory N2 neutrophils and proinflammatory platelet production and inhibits natural killers cells and CD8+ T cell production [82,83].

CAF-derived cytokines and CAF-remodeling enzymes lysyl-oxidase (LOX) and metalloproteinases (MMPs) regulate tumor immune evasion, promote growth and metastasis and can modify the tumor prognosis. In renal cancer, CAF-derived enzymes MMP 1, 9, 11 and 19 and LOXL 2 and 3 are associated with unfavorable prognostics [81].

A recent study focusing on 208 ccRCC demonstrated that a positive cytoplasmic immunostaining of FAP in the stromal fibroblasts was associated with a large tumor diameter (≥4 cm), high-grade (G3/4) tumors and high-stage (≥pT3) tumors. Patients FAP-positive had significantly shorter survival after 5, 10 and 15 years of follow-up [84].

Nowadays, no anticancer drug specifically targets CAFs, even if most of VEGFR TKI are also inhibitors of the platelet-derived growth factor (PDFG) receptor, known to regulate CAFs.

#### 3.3.3. Cancer-Associated Adipocytes (CAAs)

Being located in the retroperitoneum, kidneys are located close to fat pads. Inside the fat pads, the cancer-associated adipocytes (CAAs) are players of tumor growth. CAAs act as physical protectors of the tumor. They also contribute to the thermal factors involved in the insulation, the energy storage and the secretion of tumor invasion [85]. CAAs secrete molecules such as MMPs, collagens, fibronectin or cathepsin, which remodulate the extra cellular matrix (ECM). They are also able to induce neovascularization via the secretion of angiogenic factors such as vascular endothelial growth factor (VEGF), leptin, hepatocyte growth factor (HGF) or TGF-β. Moreover, adipose tissue is often associated with hypoxia, resulting in an upregulation of the proangiogenic signaling pathways [86,87,88]. Among the released factors, leptin is mainly known for its role in energy homeostasis but also plays an important role for the T cell-adaptive immune response [89]. In a recent publication, Campo-Verde-Arbocco et al. demonstrated that leptin released by human adipose tissue from renal cell carcinoma located near the tumor could enhance the invasive potential of renal epithelial cell lines [90].

### 3.4. PD-L1

PD-L1 can be expressed by tumor cells (TC) but, also, immune cells (IC), including myeloid cells and lymphocytes. Although the PD-L1 status is recognized as a predictive marker of response to ICI in some tumor types (notably, non-small cell lung carcinoma) [91], it does not appear to be a relevant predictive biomarker in mccRCC. In the KEYNOTE 426 [10] and the Checkmate 214 [11] studies, patients treated with ICI (Pembrolizumab+Axitinib and Nivolumab+Ipilimumab, respectively) had better overall survival (OS) compared to patients of the Sunitinib group, regardless of the PD-L1 status (PD-L1 combined a positive score superior or inferior to 1 in KEYNOTE 426 and PD-L1 expression superior or inferior to 1% in Checkmate 214). Furthermore, in the JAVELIN RENAL 101 study [92], no difference in terms of the PFS was demonstrated according to the PD-L1 status concerning patients treated with avelumab+axitinib (13.8 months for the PD-L1 expression ≥1% group and the overall population).

This may be partly explained by the difficulty to find a score reflecting the PD-L1 expression. First, it can be obtained via several scores, considering different cell types (TC or IC), each of the previously mentioned studies using its own counting method or threshold. Then, the expression of PD-L1 is heterogeneous according to the analyzed site (primary tumor or metastasis) and even within the same tumor [93]. All these data raise questions about the robustness and reproducibility of such a marker. In conclusion, there are several anti-human PD-L1 clones (most used: 22C3, 28–8 and SP142), as well as different positivity thresholds for each of the ICI studied. These different issues limit the interpretation of the PD-L1 status as a marker in response to ICI and highlight the need for a standardization of practices.

The prognostic and predictive values of the major TME components in RCC are summarized in Table 1.

## 4. TME-Related mRNA Signatures to Predict Systemic Treatment Efficacy

As mentioned above, PD-L1 expression is insufficient to assess safely which mccRCC patient would respond to ICI. The current clinical, biological and histological markers as an International Metastatic RCC Database Consortium (IMDC) score, Fuhrman grade, necrosis, vascular emboli or performance status are also imperfect to guide our therapeutic choice. The MSKCC risk score was developed during the cytokine era and the IMDC risk score during the targeted therapy era, and their accuracy may be compromised in the ICI era.

Applying the CIT (Carted Identité des Tumeurs) classification, Beuselinck et al. [96] demonstrated that favorable IMDC patients mainly belong to the ccrcc2 group, whereas intermediate and unfavorable IMDC patients were very heterogeneous. This molecular heterogeneity could explain the different patterns in response to the ICI observed among the same IMDC prognostic group. Briefly, the CIT program uses a 35-gene expression mRNA signature and is based on the unsupervised clustering of transcriptomic data from frozen tumor samples; patients with metastatic ccRCC were classified in four groups with distinct biological features: ccrcc1 = “c-myc-up”, ccrcc2 = “classical”, ccrcc3 = “normal-like” and ccrcc4 = “c-myc-up and immune-up”, representing, respectively, 33%, 41%, 11% and 15% of patients. A minimal 35-genes signature was built to classify mccRCC patients according to these four groups.

Figure 3 summarizes in a simplified way the molecular grouping according to the classifier described by Beuselinck et al. [96].

Molecular signatures using transcriptomic analysis and computational procedure with unsupervised or supervised cluster analysis are emerging as a new promising predictive biomarker in response to ICI in mccRCC.

In a metastatic context, three main genomic signatures were developed: IMmotion signature, JAVELIN Renal signature and CIT signature. They basically identified three mRNA profiles: (i) angiogenic (angio), (ii) T-effector (Teff) and the myeloid profile.

### 4.1. Angiogenesis Signature (IMmotion)

The phase II clinical trial IMmotion 150 compared atezolizumab 1200 mg every three weeks versus sunitinib (50 mg) orally once-daily for four weeks (six-week cycles) versus the combination of atezolizumab 1200 mg every three weeks and bevacizumab 15 mg/kg every three weeks in first-line mccRCC [97]. The coprimary endpoints were the PFS among the intention to treat the population and among the PD-L1+ population. The intent-to-treat PFS hazard ratios for atezolizumab+bevacizumab or atezolizumab monotherapy versus sunitinib were 1.0 (95% CI = 0.69–1.45) and 1.19 (95% CI = 0.82–1.71), respectively, and the PD-L1+ PFS hazard ratios were 0.64 (95% CI, 0.38–1.08) and 1.03 (95% CI, 0.63–1.67), respectively.

This study included an elegant exploratory biomarkers analysis to evaluate the putative predictive values of the TME. Three biological axes (the angiogenesis, the pre-existing immunity determined by the T-cell effective response, IFN gamma response, ICI expression and antigenic presentation and the immunosuppressive myeloid inflammation) were interrogated to build a predictive genomic signature. A high angiogenic (Angio^high^) signature was associated with a high vascular density, whereas a high T-effector (Teff^high^) signature was characterized by high CD8+ T cell infiltration linked with a pre-existing adaptive immunity.

Among patients receiving sunitinib, an Angio^high^ signature was associated with an objective response rate (ORR) of 46% versus 9% for an Angio^low^ signature. For patients harboring an Angio^low^ signature, PFS were, respectively, 11.4 and 3.7 months with the atezolizumab-bevacizumab combination and sunitinib. For patients harboring an Angio^high^ signature, PFS was numerically improved with sunitinib with a median PFS of 19.5 months versus 11.4 for atezolizumab-bevacizumab. No significant difference was identified between the sunitinib arm and atezolizumab-bevacizumab arm, suggesting that VEGFR TKI could remain a reasonable treatment in this selected population [97].

The phase III clinical trial IMmotion 151 [98] confirmed the prognostic and predictive values of the previously described angiogenic and T-effector signatures [99]. Eight hundred and fifty-one patients were randomized between atezolizumab 1200 mg every three weeks plus bevacizumab 15 mg/kg every three weeks and sunitinib (50 mg) orally once-daily for four weeks (six-week cycle). Within the sunitinib arm, an Angio^high^ signature was associated with an improved PFS: 10.12 months versus 5.95 months for patients with an Angio^low^ signature, HR = 0.59 (CI (confidence interval) 95% = 0.47–0.75). Within this treatment arm, the survival outcomes were not improved by the combination of atezolizumab and sunitinib (HR = 0.95, CI 95% = 0.76–1.19). On the opposite, within the Angio^low^ population, the PFS was significantly improved by the combination: 8.94 months versus 5.95 months with sunitinib monotherapy (HR = 0.68 (CI 95% = 0.52–0.88).

To summarize, an Angio^high^ signature is associated with a good prognostic without a significant survival difference between the sunitinib and atezolizumab-bevacizumab arms. This is in accordance with Hakimi’s data [100]. In the case of an Angio^low^ signature, the combination arm should be preferred.

### 4.2. Immune Signatures

#### 4.2.1. T-Effector and Myeloid Signatures (IMmotion and Javelin Renal)

Within IMmotion 150, patients receiving the atezolizumab-bevacizumab combination and harboring a Teff^high^ signature experienced improved survival outcomes compared to those harboring a Teff^low^ signature. The ORR were, respectively, 49% and 16%. The PFS were, respectively, 21.6 and 5.6 months. A high myeloid signature was associated with a pejorative PFS across all treatment arms. For patients with a high Teff and high myeloid signatures, PFS was improved with the atezolizumab-bevacizumab combination: 25 months versus seven months with sunitinib and two months with atezolizumab monotherapy, suggesting that this combination could overcome the primary resistance induced by an immunosuppressive and inflammatory TME [97].

Within IMmotion 151, the Teff^high^ population experienced a significantly improved PFS with the atezolizumab-bevacizumab versus sunitinib monotherapy with, respectively, median PFS of 12.45 versus 8.34 months (HR = 0.76 CI 95% = 0.59–0.99). No difference was observed within the Teff^low^ population [99].

In the case of a Teff^high^ signature, the combination arm should also be preferred. It should be noted that, in this phase III, no data related to the myeloid signature were reported to date.

The JAVELIN Renal study, a phase III trial, compared avelumab 10 mg/kg intravenously every two weeks plus axitinib (5 mg) orally twice-daily versus sunitinib (50 mg) orally once-daily for four weeks (six-week cycle) for advanced RCC. The PFS was significantly longer with avelumab plus axitinib than with sunitinib with, respectively, a median PFS of 13.8 months and 8.4 months (HR = 0.69, 95% CI = 0.56–0.84, *p* < 0.001) [92].

Recent outcomes from the biomarker analysis on the tumor samples from JAVELIN Renal 101 were reported by Choueiri et al. at the ASCO 2019 [101]. The transcriptomic analyses enabled the validation of a new genomic signature based on the expression of 26 genes. This innovative immune-related signature incorporates pathway indicators for TCR (T Cell Receptor) signalization, T-cell activation, proliferation and differentiation, NK-cells cytotoxicity and other immune responses, such as IFN-γ signaling. In the avelumab plus axitinib arm, a high level of expression of this signature was associated with an improved PFS: 15.2 months versus 9.8 months, *p* = 0.0019. No difference was observed in the sunitinib arm.

Very interestingly, the IMmotion signatures were evaluated in the JAVELIN Renal 101 population. An Angio^high^ signature was statistically associated with an improved PFS in the sunitinib arm. A Teff^high^ signature was numerically associated with an improved PFS in the avelumab-axitinib arm versus the sunitinib one, but statistical significance was not reached (HR = 0.79, CI 95% = 0.58–1.08 *p* = 0.14). The myeloid signature was associated with the pejorative survival outcomes, but statistical significance was not reached [101].

#### 4.2.2. Post-Hoc Analysis from the Phase III Checkmate 214

The JAVELIN renal 101, IMmotion 150 Teff and IMmotion 150 myeloid signatures were applied to the Checkmate 214 patients. For patients receiving sunitinib, a high IMmotion 150 angiogenesis score was associated with a better PFS (*p* = 0.02), but this did not extend to the OS. No association between the immune signature, myeloid signature and survival was identified. We need to be mindful that these signatures were developed in patients treated with anti-PD (L)-1 and antiangiogenic therapy, which may limit their applicability in the Checkmate 214 context. A gene set enrichment was applied to compare patients at the relative extreme of responses. Genes related to TNF-a signaling, epithelial mesenchymal transition, KRAS (Kirsten vrat sarcoma viral oncogene homolog) signaling, inflammatory response, angiogenesis, heme metabolism, TGF-β signaling and myogenesis were enriched in patients receiving the nivolumab-ipilimumab combination and harboring a PFS >18 months. On the opposite, genes related to the IFN-a response, oxidative phosphorylation, IFN gamma response, DNA repair, reactive oxygen species pathway, MYC (MYC pathway), fatty acid metabolism, adipogenesis and coagulation were enriched in patients harboring a PFS < 18 months and in patients receiving the nivolumab-ipilimumab combination [102].

The technical characteristics and predictive values of the signatures described above are summarized in Table 2.

### 4.3. Strengths and Weaknesses of Genomic Signatures

The genomic signatures offer promising perspectives, both in the adjuvant and metastatic settings. In the adjuvant setting, the aim is to identify patients with a high risk of recurrence for whom an adjuvant treatment may be beneficial. Oncotype Dx^®^, Mammaprint^®^ and Prosigna^®^ are three genomic signatures currently used in breast cancer to assess the survival benefit of an adjuvant chemotherapy in RH+ HER2-node (Human Epidermal Growth Factor Receptor-2) negative breast cancer. Disappointingly, until today, no trial has used the predictive and prognostic values of genomic signatures in RCC to identify the patients having the highest risk of recurrence and, thus, being subjected to the benefits from an anti-VEGFR TKI adjuvant treatment [104,105]. In the recent phase III randomized clinical trial KEYNOTE 564 (NCT03142334), assessing pembrolizumab versus a placebo in the ccRCC adjuvant setting, no genomic signature was used to screen the eligible patients.

Even if the genomic signatures described above seem to be the most robust predictive tool to guide our therapeutic choices, this enthusiasm should be tempered because of technical limitations. In clinical practice, tissue biopsies are classically Formalin-Fixed and Paraffin-Embedded (FFPE), which can cause cytosine deamination and artefacts. The BIONIKK study aimed to evaluate the feasibility of transcriptomic analyses of FFPE samples. Moreover, the accessibility and the reproducibility of these techniques remains a major challenge in the current practice. The intratumor heterogeneity (ITH) is also a major limitation. Using whole-exome sequencing and phylogenetic reconstruction, Swanton and Gerlinger highlighted an intratumoral heterogeneity within the primary tumor and metastatic sites and a branched evolutionary tumor growth with potential new driver mutations identified in the metastatic sites. They also reported that a single biopsy is not representative of the entire tumor bulk, as a single biopsy revealed approximately 55% of all mutations detected in the corresponding tumor. Epigenetic events may contribute to the differences in gene expression between primary and metastatic sites [106,107]. Whether this mutational profile heterogeneity led to gene expression signature heterogeneity is unknown. According to a recent study of the Mayo Clinic, genomic signature clear cell types A and B (ccA/ccB) were discordant 43% between the primary and metastatic sites. Briefly, the ccA/ccB signature, based on transcriptomic data and unsupervised clustering, was thus built by Brannon and colleagues. Two clusters were identified: one characterized by a high expression of genes involved in angiogenesis, beta oxidation, pyruvate and organic acid metabolism and the other characterized by a high expression of genes involved in cell differentiation, cell cycle, the TGF-β pathway, the Wnt-β catenin pathway and epithelial-to-mesenchymal transition. Improved progression-free survival (PFS) and overall survival were observed among the ccA subgroup [108].

A genomic signature based only on the primary tumor site is insufficient to obtain a comprehensive description of the tumor biology. The number of required biopsies remains controversial. Single-cell analysis and mass cytometry with an extensive antibody panel are other emerging study methods offering promising results. Using high-dimensional single-cell mass cytometry and the bioinformatics pipeline, Krieg et al. demonstrated that the frequency of CD14^+^CD16^−^HLA-DR^hi^ monocytes was predictive of PFS and OS for melanoma patients receiving anti-PD-1. They postulated that the frequency of monocytes may be used to support the therapeutic choice [109]. In their kidney cancer immune atlas, Chevrier et al. identified several dozen immune cell populations among the 17 TAM phenotypes and 22 T cell phenotypes. CD38^+^CD204^+^CD206^−^ TAM was identified as a poor prognosis factor in this series of primary tumors [14].

## 5. Perspectives: The BIONIKK Trial as an Example of Integrative TME Analyses

BIONIKK (NCT02960906) was the first prospective clinical trial studying the personalization of treatments according to tumor molecular characteristics in mccRCC [110,111]. Molecular characteristics are assessed by the CIT classification, described in Section 4.

Interestingly, the CIT signature was able to strongly predict the response to a first-line VEGFR TKI. On the whole population, nonresponders mainly belonged to the ccrcc1 and four groups: 22% of progressive disease (PD) in the ccrcc1 group and 27% in the ccrcc4 group versus, respectively, 3% and 2% in the ccrcc2 and -3 groups. Responders mainly belonged to the ccrcc2 and -3 groups (respectively, 53% and 70% of the partial response (PR) and complete response (CR) versus, respectively, 41% and 21% of the PR and CR in the ccrcc1 and -4 groups, *p* = 0.005). Similar results were observed regarding the OS and the PFS. The OS were, respectively, 24, 14, 35 and 50 months (*p* = 0.001) in ccrcc 1, -4, -2 and -3. The PFS were, respectively, 13, 8, 19 and 24 months (*p* = 0.001) in ccrcc1, -4, -2 and -3. Interestingly, in the multivariate analysis, the CIT signature was the only factor significantly associated with the survival outcomes.

Those results could be explained by a heterogenous composition of the TME between the four subgroups: ccrcc4 are characterized by a strong inflammatory Th1-oriented but immunologically suppressed microenvironment, whereas ccrcc1 tumors are characterized by a very low T-cell infiltration and could be summarized as “cold” tumors or “immune desert” tumors. The ccrcc2 subtype was not characterized by specific pathways and showed an intermediate expression signature between the ccrcc3 and ccrcc1/crcc4-related profiles. It seemed associated with a high angiogenic signature [111].

Based on these results, we launched the BIONIKK trial with the following hypotheses: ccrcc4 tumors should respond well to nivolumab alone given their high T-cell infiltration, whereas ccrcc1 tumors may need the addition of ipilimumab or nivolumab to prime and to attract effector T cells in the core of the tumor. As ccrcc2 and -3 tumors were responsive to sunitinib, we hypothesized that TKI would be very efficient in a prospective trial.

BIONIKK was a French multicentric molecular-driven randomized phase 2 trial (NCT02960906) where mccRCC patients were randomized to receive the first-line treatment according to their molecular group defined by the 35-gene classifier. Patients in groups 1 and 4 were randomized to receive nivolumab alone (arms 1A and 4A) or nivolumab plus ipilimumab for four injections followed by nivolumab alone (arms 1B and 4B). Patients in groups 2 and 3 were randomized to receive nivolumab plus ipilimumab followed by nivolumab alone (arms 2B and 3B) or a tyrosine kinase inhibitor (sunitinib or pazopanib at the investigator’s choice (arms 2C and 3C)). The main objective was the overall response rates by the treatment arm and molecular group. The main results were reported at the last ESMO (virtual) meeting and confirmed the hypotheses describe above: nivolumab provided a comparable ORR to nivolumab-ipilimumab in ccrcc4 but not in ccrcc1, whereas the ORR were comparable between the TKI and nivolumab-ipilimumab in ccrcc2 [110]. A huge biomarker program is ongoing, including the evaluation of the mRNA signatures described earlier in this review.

## 6. Conclusions

Within a few years, the therapeutic armamentarium of mccRCC has changed dramatically with the emergence of targeted therapy (especially anti-VEGFR TKI) and ICI, used alone or as a combination either with other ICI or with anti-VEGFR TKI. This was made possible by a greater understanding of the tumor biology, supported by the development of innovative TME study methods, such as multiplex IHC, flow cytometry, transcriptomic data and deconvolution tools. Despite these major technical and therapeutic advances, mccRCC still have a poor prognosis, with a median five-year survival rate lower than 10%, and a significant portion of patients experience primary or secondary resistance. Current clinical and biological markers, such as PD-L1 expression, tumor mutational burden or MSKCC scores, fail to predict ICI and ICI-antiangiogenic combination efficacy. Among the new predictive markers, mRNA signatures appear the most promising. In this context, the first results of the molecular-driven phase 2 trial BIONIKK show that prospective molecular selection is feasible and enables to enrich the response rate in patients treated with TKI or ICI alone or in combination. The ancillary program from the BIONIKK trial could inform more precisely on the optimal biomarkers to use to adapt treatments in the first-line setting of mccRCC.

## Figures and Tables

**Figure 1 cancers-13-00231-f001:**
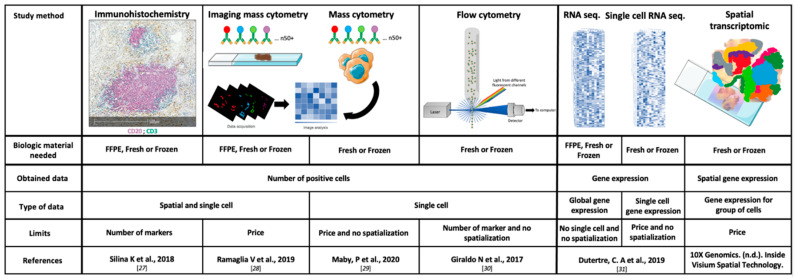
Technical characteristics of the main study methods of the tumor microenvironment. FFPE: Formalin-Fixed Paraffin-Embedded and RNA: ribonucleic acid.

**Figure 2 cancers-13-00231-f002:**
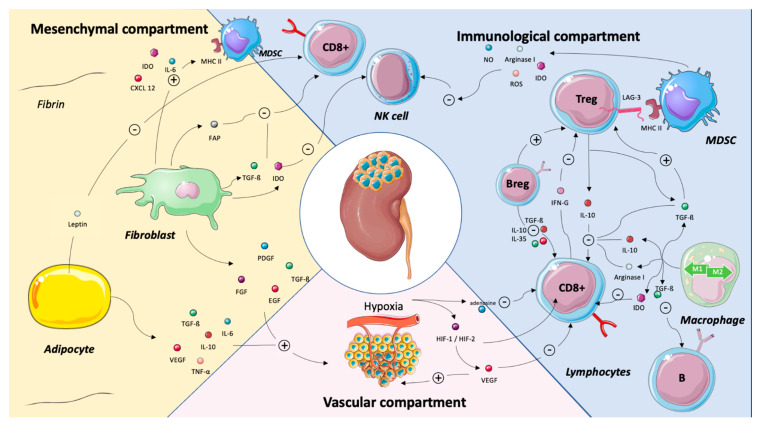
Major cross-talks between the mesenchymal, the immune and the vascular compartments in renal cell carcinoma. Abbreviations: B-reg: B regulatory cells, CD8: cluster of differentiation 8, CXCL2: chemokine (C-X-C motif), EGF: epidermal growth factor, FAP: fibroblast activation protein, FGF: fibroblast growth factor, HIF-1/HIF-2: hypoxia-induced factor-1/hypoxia)-induced factor 2, IDO: indoleamine 2,3-dioxygenase, IFN-γ: interferon γ, IL: interleukin, LAG3: lymphocyte-activation gene 3, MHC: major histocompatibility complex, MDSC: myeloid-derived suppressive cells, NK: natural killer, NO: nitric oxide, PDGF: platelet-derived growth factor, ROS: reactive oxygen species, TGF-β: transforming growth factor beta, TNF-α: tumor necrosis factor alpha, T-reg: T-regulatory cells and VEGF: vascular endothelial growth factor. Legend: The tumor microenvironment is a complex and dynamic network composed both of tumor cells, adaptive and immune cells, endothelial cells and mesenchymal cells as adipocytes and cancer-associated fibroblasts. Structural molecules and extra cellular matrix shape this network. This illustration is not intended to be comprehensive but, rather, to highlight key cross-talks between the immune, the vascular and the mesenchymal compartments. Adipocytes favor tumor progression by inhibiting CD8+ T cells via the leptin release and by stimulating angiogenesis via the release of IL-6, Il-10, TGFB, VEGF or TNF-a. By secreting IDO, IL-6, FAP, TGF-β and IDO, the fibroblasts stimulate MDSC and inhibit CD8 + T cells and NK cells. They also stimulate angiogenesis. The VEGF released by the vascular compartment of renal cell carcinoma has an immunosuppressive effect by inhibiting CD8+ T cells. The interactions between the immune cells are manifold. Basically, FoxP3+ T cells inhibit NK cells, CD8+ T cells and favor macrophage type 2 polarization. B-reg cells stimulate FoxP3+ T cells and inhibit CD8+ T cells. Depending on their polarization, tumor-associated macrophages have pro- or antitumor effects.

**Figure 3 cancers-13-00231-f003:**
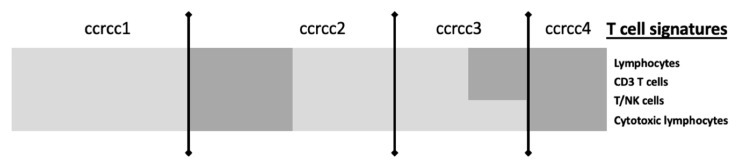
Simplified view of T-cell signatures according to the ccRCC molecular subgroup, adapted from The Human Tumor Microenvironment, Vano et al. Oncoimmunology 2018 [95]. Legend: ccrcc molecular subgroups have different gene expression immune profiles. Ccrcc 1 are immune-desert, ccrcc 4 are immune-high, ccrcc3 are immune-competent and ccrcc 2 immune-mixed. Light grey means Underexpression; Dark grey means Overexpression.

**Table 1 cancers-13-00231-t001:** Prognostic and predictive value of the major TME components in RCC.

TME Element	Status	Associated Prognostic in RCC	Predictive Value for Response to ICI in ccRCC
Cells			
CD8+ T cells [37,38].	High density	Poor	Insufficient data
Regulatory CD4+ T cells [54]	High density	Poor	No
Tumor-associated Macrophages [50]	High density	Poor	Insufficient data
B cells [61]	High density	Good	Insufficient data
Tertiary Lymphoid Structure [56,61]	High density	Good	Insufficient data
Immune checkpoints			
LAG3 [42,43]	Overexpression	Poor	Insufficient data
TIM3 [42,43]	Overexpression	Poor *	Insufficient data
PD-L1 [9,10,91,94,95]	Overexpression	Poor	No **

Legends: * Tim-3 Expression was associated with poor clinical outcome in RCC when co-expressed with PD-1+ and CD8+ on T Cells. ** Discordant data have been identified in Checkmate 214: PD-L1 expression ≥ 1% was associated with a better outcomes with nivolumab-ipilimumab but superiority of this combination over sunitinib is maintained in PD-L1 < 1%. Abbreviations: ICI: Immune Checkpoint Inhibitors. ccRCC: clear cell renal cell carcinoma. LAG3: lymphocyte-activation gene 3. TIM3: T-cell immunoglobulin and mucin domain-containing protein-3.

**Table 2 cancers-13-00231-t002:** Main technical characteristics and predictive values of the responses of the three major signatures evaluated in metastatic renal cell carcinoma. RT-qPCR: reverse-transcriptase quantitative PCR, IL: interleukin, TGF-β: transforming growth factor β, PFS: progression-free survival and FFPE: Formalin-Fixed Paraffin-Embedded.

Signatures	Study Design	Number of Patients	Genes Involved in the Signature	Treatments	Biological Material Needed	Study Method	Predictive Value of Response	
							TKI	ICI
CIT: classification ccrcc 1-2-3-4 Beuselinck et al. [93]	Retrospective study	53 (exploratory cohort) 47 (validation cohort)	Inflammation, myeloid activation, myeloid cells migration, Th1/ Th2 polarization, T cell, CMH I, TGFb, IL10, IL17	Sunitinib	Frozen samples	micro-array (exploratory cohort) RT-qPCR (validation cohort)	YES *improved ORR, PFS and OS for ccrcc2 et 3 groups*	On going (BIONIKK phase II clinical trial NCT 02960906)
IMmotion 150 McDermott et al. [62] IMmotion 151 Rini et al. [103]	Randomized phase II and phase III prospective studies	300 (IMmotion 150) 851 (IMmotion 151)	Angiogenesis, immune response, IFNg, inflammation, myeloid cells	Atezolizumab-bevacizumab vs sunitinib (Atezolizumab-bevacizumab vs atezolizumab pour la phase II)	FFPE samples	RNAseq	YES Improved PFS with sunitinib for Angio^high^	YES Improved PFS with atezolizumab-bevacizumab for Angio^low^ et Teff^high^
JAVELIN Renal 101 Choueiri et al. [96]	Randomized phase III prospective study	886	Immune response (TcR signalisation, activation-proliferation and T cells differentiation), chimiokines, NK	Avelumab-axitinib vs sunitinib	FFPE samples	RNAseq	NO	YES Improved PFS with avelumab-axitinib for pts with high expression

## Data Availability

Not applicable.

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
