# Peer review of "Tumor Microenvironment Features as Predictive Biomarkers of Response to Immune Checkpoint Inhibitors (ICI) in Metastatic Clear Cell Renal Cell Carcinoma (mccRCC)"

_cancers, 2021, doi:10.3390/cancers13020231_

Round 1

Reviewer 1 Report

This manuscript provides discussion of the role of tumor microenvironment and its transcriptomic profiles in prediction of response of clear cell-RCC to immune checkpoint inhibitors (ICI) and/or anti-angiogenic therapies. Unlike in other cancers, PD-L1 expression and tumor mutational burden do not seem to be predictive of clear cell-RCC response to the ICI, which indicates the need for development of other predictive biomarkers.  

Comments:

Overarching: The manuscript would benefit from substantial editing of language and style. Example: Page 2 Line3...: too long and borderline incomprehensible sentence. 

Specific:

Page 3: Brief comparison of the efficacy of cytokine therapies vs. targeted therapies vs. immune checkpoint-directed therapies of clear cell RCC would be helpful as a part of the introduction.

Page 3 Section 2.2.3 "These devices..." The discussion is about a software for deconvolution of transcriptomic data to estimate abundance of immune cells and some other cell types. Instead of calling it "devices", it would be better to characterize it as a software, and provide some details, such as what is the input (gene expression data from which platforms?) and what is the output (proportions  of cell types or absolute quantities of cell types in arbitrary units...). 

Page 3: "28 subclasses of immune cells". MCP Counter does not seem to attribute gene expression data to 28 cell types and it seems that the correct number of cell types is 10 (8 immune cell types + endothelial cells + fibroblasts). 

Page 3: "T cell infiltration and the expression level of MHC class I were associated with PD-1 efficacy [21,22]." Please explain what is meant by PD-1 efficacy: Is it efficacy of PD-1 directed therapies, or expression of PD-1, or activity status of PD-1 pathway? 

Figure 2: Readability of the figure legend would improve, if the abbreviations were shown in a separate list of abbreviations. Some abbreviations are not explained (e.g., "Breg"). "FoxP3+" should be for consistency called "Treg"; IDO is shown as if it were released from "FoxP3+" which is not correct, and IDO is not clearly shown as released by tumor associated fibroblasts and MDSC. 

Page 7: "Focusing on RCC, this negative correlation could be explained in part by the high levels of CTLA and PD-1 expression on T cells [22]."

The biological basis of the implied difference in expression profiles of T-cells among RCC and other cancers should be discussed in more detail.  

Page 7: "Granier et al. showed that the co-expression of TIM-3 and PD-L1 on CD8 T cells was correlated with higher TNM stage"

Should be PD-1 instead of PD-L1

Page 11:"Beuselinck et al. [93] demonstrated that favorable IMDC patients mainly belong to the ccrcc2 group whereas intermediate and unfavorable IMDC patients were very heterogeneous"

The CIT classification outcome is mentioned here before this concept and classification groups were introduced later in page 15. This should be fixed, because readers should be able to read the text continuously without need to seeking necessary explanations later in the text or in other sources. 

Page 12: "No significant difference was identified between the different arms, suggesting that VEGFR TKI could be a reasonable treatment in this selected population[94]."

Difference in what? This should be stated clearly. 

Page 15: This should read "HR+HER2-" instead of "RH+HER2-"

Page 15: "Transcriptomic analysis requires frozen tissue, to preserve the nucleic acids."

Freezing tissue is not a necessary condition. The tissue preserved by RNA preserving solutions can be stored for some time refrigerated or even at room temperature for shorter periods of time. 

Page 15: ccA/ccB is just mentioned (in abbreviation) without corroboration. This clear cell RCC classification based on ClearCode34 gene signature should be briefly introduced, including its significance; otherwise, the information would be too fragmented. 

Page 15: "Using whole exome sequencing and phylogenetic reconstruction, Swanton and Gerlinger have demonstrated that the majority of the mutations detected on the primary tumor where not shared by the metastatic sites [102,103].Whether this mutational profile heterogeneity lead to gene expression signature heterogeneity is unknown."

Is the point here that majority of mutations found in primary tumors are lost in metastases? It would seem more likely that mutations in metastatic sites were new and not found in primary tumor sites, but the statement in the manuscript implies otherwise.

In addition, differences in gene expression between primary and metastatic sites can be attributed to many epigenetic events that are more likely to be responsible for the changes in gene expression rather than changes in DNA sequence of tumor cells between primary and metastatic sites. 

Page 15: "A minimal 35-genes signature was built to classify mccRCC patients according to these 4 groups"

This reviewer recommends that clustering in the form of a heatmap is presented for these 35 genes and selection of RCC representative of the 4 groups, so that readers can correlate group membership of specific cancers specimens with expression of specific genes. 

Page 15: CR instead of RC

Author Response

AUTHOR RESPONSE LETTER

Manuscript reference number: cancers-1011754

Dear Dr. Paul Reynolds and Mr. Grant D Stewart

Thank you for the review of our manuscript, entitled: “Tumor microenvironment features as predictive biomarkers of response to immune checkpoint inhibitors (ICI) in metastatic clear cell renal cell carcinoma (mccRCC)”. We sincerely appreciate all valuable comments and suggestions, which helped us to improve the quality of our manuscript. Our responses to the Reviewers’ comments are described below in a point-by-point manner. Appropriated changes, suggested by the Reviewers, have been added to the manuscript and all changes are in red colored text in the manuscript.

We hereby affirm that this manuscript has not been published previously, accepted for publication elsewhere and is not under consideration for publication elsewhere.

We hope that this clarification may enhance the quality of our paper to the level required for publication in Cancers.

We look forward to hearing from you

Yours sincerely,

Dr Yann -Alexandre VANO, MD PhD

Dr Audrey SIMONAGGIO, MD

Responses to the Reviewers’ specific comments:

Reviewer #1

We would like to thank Reviewers #1 for taking the time and effort necessary to review the manuscript. We sincerely appreciate all valuable comments and suggestions, helping us to clarify some key points and improving the quality of the manuscript.

Comments to authors:

This manuscript provides discussion of the role of tumor microenvironment and its transcriptomic profiles in prediction of response of clear cell-RCC to immune checkpoint inhibitors (ICI) and/or anti-angiogenic therapies. Unlike in other cancers, PD-L1 expression and tumor mutational burden do not seem to be predictive of clear cell-RCC response to the ICI, which indicates the need for development of other predictive biomarkers.  

Comments:

1.Overarching: The manuscript would benefit from substantial editing of language and style. Example: Page 2 Line3...: too long and borderline incomprehensible sentence. 

Response: A careful proofread has been done to improve English language and style. Appropriated changes have been added to the manuscript.

For example, the following sentence has been modified accordingly: “Consequently during the 2000s, anti-VEGFR TKI and mTOR inhibitors emerged as the new standards of care for mccRCC.  Sunitinib [5] and pazopanib [6] were standard of care for untreated Memorial Sloan Kettering Cancer Center (MSKCC) favorable and intermediate group. Temsirolimus and everolimus [7] were recommended for first line MSKCC intermediate and poor risk mccRCC patients and after anti-VEGFR TKI failure »

Specific:

  1. Page 3: Brief comparison of the efficacy of cytokine therapies vs. targeted therapies vs. immune checkpoint-directed therapies of clear cell RCC would be helpful as a part of the introduction.

Response: We thank the reviewer for this important comment. Appropriate changes have been made to clarify the introduction.

Following sentences have been added related to the efficacy of cytokine therapies and targeted therapies:

“Objective response rates were low (5-20%) and safety profile was often limiting because of cardiac and respiratory toxic effects. Interferon alpha gave an improvement in 1-year survival of 12% and in median survival of 2.5 months versus medroxyprogesterone acetate.”

“Compared with placebo or interferon, Sunitinib and pazopanib provided a survival benefit with a median overall survival of 28.4 months in the pazopanib group and 29.3 months in the sunitinib group”.

“Survival benefit offered with these new combinations will be extensively described in Section 4”.

  1. Page 3 Section 2.2.3 "These devices..." The discussion is about a software for deconvolution of transcriptomic data to estimate abundance of immune cells and some other cell types. Instead of calling it "devices", it would be better to characterize it as a software, and provide some details, such as what is the input (gene expression data from which platforms?) and what is the output (proportions  of cell types or absolute quantities of cell types in arbitrary units...). 

Response : We thank the reviewer for this important  comment. Substantial modifications have been added to improve the comprehension of this section :

“ The Microenvironment Cell Population-counter (MCP counter) is a software which uses  transcriptomic data and deconvolution tools to “decipher the contribution of different cell populations to the overall transcriptomic signal of heterogeneous tissue samples” [20]. Deconvolution is an algorithm-based method used to deduce abundance of cell types and cell expression into an heterogeneous sample, using gene expression data [20]. MCP counter allows the quantification of the absolute abundance of 10 distinct populations (8 immune cell types  including T cells, CD8+ T cells, NK cells, cytotoxic lymphocytes, B cell lineage, monocytic lineage cells, myeloid dendritic cells and neutrophils and 2 non-immune stromal populations including endothelial cells and fibroblasts [20]”. 

  1. Page 3: "28 subclasses of immune cells". MCP Counter does not seem to attribute gene expression data to 28 cell types and it seems that the correct number of cell types is 10 (8 immune cell types + endothelial cells + fibroblasts). 

Response: We thank the reviewer for this essential comment. Indeed, MCP counter attributes gene expression data to 10 distinct populations (8 immune cell types (T cells, CD8+ T cells, NK cells, cytotoxic lymphocytes, B cell lineage, monocytic lineage cells, myeloid dendritic cells and neutrophils) and two non-immune stromal populations (endothelial cells + fibroblasts). Appropriate modifications have been done.

  1. Page 3: "T cell infiltration and the expression level of MHC class I were associated with PD-1 efficacy [21,22]." Please explain what is meant by PD-1 efficacy: Is it efficacy of PD-1 directed therapies, or expression of PD-1, or activity status of PD-1 pathway? 

Response: PD-1 efficacy means efficacy of anti-PD-1 nivolumab.  Appropriate comments have been added to clarify this section: “focusing on a small sample of patients (n=6), the MHC class I antigen presenting machinery expression and the T cell infiltration were elevated in patients with partial or complete response to nivolumab ».

  1. Figure 2: Readability of the figure legend would improve, if the abbreviations were shown in a separate list of abbreviations. Some abbreviations are not explained (e.g., "Breg"). "FoxP3+" should be for consistency called "Treg"; IDO is shown as if it were released from "FoxP3+" which is not correct, and IDO is not clearly shown as released by tumor associated fibroblasts and MDSC. 

Response : According to this comment, appropriate changes have been done to clarify the figure 2 and improve the readability. 

Following modifications have been done:

            -explanation of all the abbreviations in a separate list.

            -modifications of the IDO release

            -substitution of “FoxP3+” by “Treg”

  1. 7. Page 7: "Focusing on RCC, this negative correlation could be explained in part by the high levels of CTLA and PD-1 expression on T cells [22]."

The biological basis of the implied difference in expression profiles of T-cells among RCC and other cancers should be discussed in more detail.  

Response : We thank the reviewer for this important issue. Some biological explanations have been added to discuss this point, as follows.

 “ Giraldo et al. identified two groups of ccRCC tumors with high CD8+ Tcell infiltrates and opposite survival outcomes. First group associated with good prognosis was characterized by high expression of immune checkpoints and no mature dendritic cells  whereas the second group associated with worse prognosis had  low expression of immune checkpoints and mature peritumoral dendritic cells. One hypothesis to explain this observation may be that CD8+ T cells are often exhausted in ccRCC and that CD8+ Tcells could only be educated in presence of high density of TLS-DC.   “

  1. Page 7: "Granier et al. showed that the co-expression of TIM-3 and PD-L1 on CD8 T cells was correlated with higher TNM stage"

Should be PD-1 instead of PD-L1

Response: PD-L1 has been written instead of PD-1. Appropriate modifications have been done.

9.Page 11:"Beuselinck et al. [93] demonstrated that favorable IMDC patients mainly belong to the ccrcc2 group whereas intermediate and unfavorable IMDC patients were very heterogeneous". The CIT classification outcome is mentioned here before this concept and classification groups were introduced later in page 15. This should be fixed, because readers should be able to read the text continuously without need to seeking necessary explanations later in the text or in other sources. 

Response: We thank the reviewer for this practical comment which enable to improve the quality of our review.

For a better understanding, the description of the CIT classification has been included into section 4 instead of section 5.

Following sentences has been added : “Briefly, The CIT program uses a 35-gene expression mRNA signature and is based on unsupervised clustering of transcriptomic data from frozen tumor samples, patients with metastatic ccRCC were classified in 4 groups with distinct biological features, ccrcc1 = “c-myc-up”,  ccrcc2 = “classical,” ccrcc3 = “normal-like” and ccrcc4 = “c-myc-up and immune-up”, representing respectively 33, 41, 11 and 15% of patients. A minimal 35-genes signature was built to classify mccRCC patients according to these 4 groups”.

  1. Page 12: "No significant difference was identified between the different arms, suggesting that VEGFR TKI could be a reasonable treatment in this selected population[94]."

Difference in what? This should be stated clearly. 

Response:  Following modifications have been added, to clarify the sentence : “For patients harboring an Angiohigh signature, PFS was numerically improved with sunitinib with a median PFS of 19.5 months versus 11.4 for atezolizumab-bevacizumab. No significant difference was identified between sunitinib arm and atezolizumab-bevacizumab arm, suggesting that VEGFR TKI could remain a reasonable treatment in this selected population”

  1. Page 15: This should read "HR+HER2-" instead of "RH+HER2-"

Response: Appropriate modifications have been done.

  1. Page 15: "Transcriptomic analysis requires frozen tissue, to preserve the nucleic acids."

Freezing tissue is not a necessary condition. The tissue preserved by RNA preserving solutions can be stored for some time refrigerated or even at room temperature for shorter periods of time. 

Response: We thank the reviewer for this important practical comment. We agree with this comment and thus following sentence: “Transcriptomic analysis requires frozen tissue, to preserve the nucleic acids” has been removed from the manuscript

  1. Page 15: ccA/ccB is just mentioned (in abbreviation) without corroboration. This clear cell RCC classification based on ClearCode34 gene signature should be briefly introduced, including its significance; otherwise, the information would be too fragmented. 

Response : A short description of the clear cell type A and B genomic signature (ccA/ccB) has been added to clarify this section:  “Briefly, the ccA/ccB signature, based on transcriptomic data and unsupervised clustering was thus built by Brannon and colleagues. Two clusters were identified: one characterized by a high expression of genes involved in angiogenesis, Beta oxidation, pyruvate and organic acids metabolism and the other characterized by a high expression of genes involved in cell differentiation, cell cycle, TGFb pathway, Wnt-Bcatenin pathway and epithelial to mesenchymal transition. Improved progression free survival (PFS) and overall survival were observed among ccA subgroup”.

14. Page 15: "Using whole exome sequencing and phylogenetic reconstruction, Swanton and Gerlinger have demonstrated that the majority of the mutations detected on the primary tumor where not shared by the metastatic sites [102,103].Whether this mutational profile heterogeneity lead to gene expression signature heterogeneity is unknown."

Is the point here that majority of mutations found in primary tumors are lost in metastases? It would seem more likely that mutations in metastatic sites were new and not found in primary tumor sites, but the statement in the manuscript implies otherwise.

In addition, differences in gene expression between primary and metastatic sites can be attributed to many epigenetic events that are more likely to be responsible for the changes in gene expression rather than changes in DNA sequence of tumor cells between primary and metastatic sites. 

Response : We totally agree that our sentence can lead to confusion. The key points revealed by Swanton and Gerlinger were the intratumoral heterogeneity within primary tumors and metastatic sites (meaning that a single biopsy is not representative of the entire tumor bulk) and the branched evolutionary tumor growth with potential new driver mutations identified in metastatic sites.

To clarify this point, following modifications have been done :

“Using whole exome sequencing and phylogenetic reconstruction, Swanton and Gerlinger have highlighed an intratumoral heterogeneity within primary tumor and metastatic sites and a branched evolutionary tumor growth with potential new driver mutations identified in metastatic sites. They also reported that a single biopsy is not representative of the entire tumor bulk as a single biopsy revealed approximately 55% of all mutations detected in the corresponding tumor. Epigenetics events may contribute to the differences in gene expression between primary and metastatic sites  [107,108]. »

15. Page 15: "A minimal 35-genes signature was built to classify mccRCC patients according to these 4 groups".

This reviewer recommends that clustering in the form of a heatmap is presented for these 35 genes and selection of RCC representative of the 4 groups, so that readers can correlate group membership of specific cancers specimens with expression of specific genes. 

Response: To the best of our knowledges, such heatmap does not exist. However, we propose to add Figure 3, adapted from The Human Tumor Microenvironment, Vano et al. Oncoimmunology 2018. and focusing on T cell signatures. This figure summarizes the molecular grouping according to the 35-gene classifier described by Beuselinck et al.

Figure 3 has been added in the manuscript.

Title . Simplified view of T cell signatures according to the ccRCC molecular subgroup.

Legend: ccrcc molecular subgroups have different gene expression immune profiles. Ccrcc 1 are immune-desert, ccrcc 4 are immune-high, ccrcc3 are immune competent and ccrcc 2 immune-mixed.

  1. Page 15: CR instead of RC

Response : Appropriate change has been made.

Reviewer 2 Report

This is a nice comprehensive review of tumor microenvironment of metastatic renal cell carcinoma.

Author Response

Reviewer #2:

We would like to thank Reviewer #2 for taking the time to review the manuscript.

Comments to authors:

Reviewer#2 did not require any revision.

Reviewer 3 Report

This is a detailed review focused on mccRCC and heterogeneity of response to SoC, the poor predictive value of traditional ICI biomarkers and the need for improved predictive biomarkers of response that take into account the complex mccRCC TME.

The mccRCC TME is discussed in detail and in particular the influence of heterotypic cell signalling and extent of immune cell infiltration on likely response to different SoC options. The authors discuss use of predictive biomarker TME signatures  (e.g. use of mRNA signatures, other multiplex signatures) for improved patient stratification.

A few minor revisions would improve readability:-

The review would benefit from a careful proofread by a native speaker of English and also checking for typological errors as it is a little 'awkward' in places e.g. 'rising of therapeutic options' replace with 'an increase in'. Replace resistances with resistance, others cells replace with other cells, cytokines therapies  replace with cytopkine theraries etc.

Section 2.2.1 'Antibodies DNA-barcoding' is poorly/not explained.

Section 2.2.3  'These devices' - these devices are not introduced, unclear what the author is referring to.

Introduction - 120,000 patients died from kidney cancer - when? 

2000's should read 2000s

Author Response

Reviewer #3:

We would like to thank Reviewer #3 for the comments and assessments. We sincerely appreciate all valuable comments helping us to clarify some key points.

Comments to authors:

This is a detailed review focused on mccRCC and heterogeneity of response to SoC, the poor predictive value of traditional ICI biomarkers and the need for improved predictive biomarkers of response that take into account the complex mccRCC TME.

The mccRCC TME is discussed in detail and in particular the influence of heterotypic cell signalling and extent of immune cell infiltration on likely response to different SoC options. The authors discuss use of predictive biomarker TME signatures  (e.g. use of mRNA signatures, other multiplex signatures) for improved patient stratification.

A few minor revisions would improve readability:

  1. The review would benefit from a careful proofread by a native speaker of English and also checking for typological errors as it is a little 'awkward' in places e.g. 'rising of therapeutic options' replace with 'an increase in'. Replace resistances with resistance, others cells replace with other cells, cytokines therapies  replace with cytopkine theraries etc.

Response. We thank the reviewer for this important comment. A careful proofread has been done to improve English language and style. Appropriated changes have been added to the manuscript and all changes are in red colored text in the manuscript.

  1. Section 2.2.1 'Antibodies DNA-barcoding' is poorly/not explained.

Response.  We thank the reviewer for this important and practical comment. According to the Reviewer’s comment, a description of the “antibodies DNA-barcoding” technique has been added to improve the comprehension of this section.

The following sentence has been modified accordingly: “Antibodies DNA-barcoding (InSituPlexâ Technology by ULTIVUE for 12-plex or CODEXâ by AKOYA Bioscience for a 29-plex maximum) is a new method of cancer cell profiling using DNA-conjugated antibodies. The DNA-conjugated antibodies contain a photo-cleavable linker which enables their release after exposure to ultraviolet light. The transmitted signal is then measured and translated to protein expression. It enables to increase the number of available markers. »

  1. Section 2.2.3 ‘These devices' - these devices are not introduced, unclear what the author is referring to.

Response. We thank the reviewer for this practical comment. Appropriate changes have been made to clarify the section.

The following sentence has been modified accordingly : “ The Microenvironment Cell Population-counter (MCP counter) is a software which uses  transcriptomic data and deconvolution tools to “decipher the contribution of different cell populations to the overall transcriptomic signal of heterogeneous tissue samples” [20]. Deconvolution is an algorithm-based method used to deduce abundance of cell types and cell expression into an heterogeneous sample, using gene expression data [20]. MCP counter allows the quantification of the absolute abundance of 10 distinct populations (8 immune cell types  including T cells, CD8+ T cells, NK cells, cytotoxic lymphocytes, B cell lineage, monocytic lineage cells, myeloid dendritic cells and neutrophils and 2 non-immune stromal populations including endothelial cells and fibroblasts [20]”. 

  1. Introduction - 120,000 patients died from kidney cancer - when? 

Response. The following sentence  has been modified accordingly : “In 2018, about 330,000 new cases were diagnosed and 120,000 patients died from kidney cancers [1].

  1. 2000's should read 2000s

Response. Appropriate change has been made.